# PROCBENCH: BENCHMARK FOR MULTI-STEP REASONING AND FOLLOWING PROCEDURE

## ABSTRACT

Reasoning is central to a wide range of intellectual activities, and while the capabilities of large language models (LLMs) continue to advance, their performance in reasoning tasks remains limited. The processes and mechanisms underlying reasoning are not yet fully understood, but key elements include path exploration, selection of relevant knowledge, and multi-step inference. Problems are solved through the synthesis of these components. In this paper, we propose a benchmark that focuses on a specific aspect of reasoning ability: the direct evaluation of multi-step inference. To this end, we design a special reasoning task where multi-step inference is specifically focused by largely eliminating path exploration and implicit knowledge utilization. Our dataset comprises pairs of explicit instructions and corresponding questions, where the procedures necessary for solving the questions are entirely detailed within the instructions. This setup allows models to solve problems solely by following the provided directives. By constructing problems that require varying numbers of steps to solve and evaluating responses at each step, we enable a thorough assessment of state-of-the-art LLMs' ability to follow instructions. To ensure the robustness of our evaluation, we include multiple distinct tasks. Furthermore, by comparing accuracy across tasks, utilizing step-aware metrics, and applying separately defined measures of complexity, we conduct experiments that offer insights into the capabilities and limitations of LLMs in reasoning tasks. Our findings have significant implications for the development of LLMs and highlight areas for future research in advancing their reasoning abilities.

## 1 INTRODUCTION

Reasoning is a fundamental component of intelligence, involving complex processes where the application of knowledge and logical inference are deeply intertwined (Wason & Johnson-Laird, 1972; Huang & Chang, 2023; Fagin et al., 1995). We define reasoning as the progression toward a specific goal through multiple steps of inference to derive new knowledge from existing information (Yu et al., 2024); it begins with goal setting, which can be self-initiated or explicitly provided, as is often the case in problem-solving; then, a series of inference is repeated until the goal is achieved, handling both explicit and implicit knowledge such as common sense or domain-specific information.

There are three classes of inference: induction, deduction and abduction (Peirce, 1992); induction is a process of generalization from a specific observation, while deduction is the opposite: from general to specific, and abduction is the inference to the best explanation from observations (Huang & Chang, 2023; Yu et al., 2024). All of them in common often require an exploration in a vast search space to determine the correct path to the goal, choosing necessary knowledge for the decisions from substantial knowledge.

Although reasoning involves such complex processes, here we focus on the process to follow a fixed path to a given goal with explicit knowledge, proposing **ProcBench**, which consists of tasks that do not require complex knowledge but can be solved by following the provided procedures. The goal of this dataset is to evaluate the ability of AI systems to follow and execute specific instructions, which we refer to as **instruction followability**. While trivial for humans, it can be challenging for AI systems that do not strictly adhere to the instructions. In choosing the tasks, we take into consideration that they have the following properties:

- The procedure to reach the goal is explicitly provided, so no search for the correct path is necessary.

- Minimal implicit knowledge beyond basic language comprehension is required to execute the procedure.

- The steps in the procedure are straightforward for humans to execute.

Numerous benchmarks (Saxton et al., 2019; Hendrycks et al., 2021; 2020a; Yu et al., 2023; Cobbe et al., 2021; Dinh et al., 2024; Suzgun et al., 2022) have been proposed to evaluate reasoning in AI systems, ranging from basic arithmetic to advanced theorem proving and competitive programming challenges (Jain et al., 2024; Lu et al., 2021; Zhuo et al., 2024). While these benchmarks have evolved to tackle more complex tasks, they often require implicit knowledge, making it difficult to isolate and assess an AI's procedural adherence. Furthermore, traditional AI evaluation methods have largely focused on final outputs, often at the expense of the reasoning process itself. This oversight can lead to systems that perform well in simple scenarios but fail when confronted with complex tasks that require careful, multi-step reasoning. ProcBench sets itself apart by emphasizing evaluative tasks that require minimal prerequisite knowledge and demanding exact adherence to instructions, whilst all necessary information is provided within the task description – thereby filling a critical gap in current evaluation methods.

Instruction followability is crucial across several key areas in AI, including reasoning (Yu et al., 2024), explainable AI (Arrieta et al., 2019), mitigating hallucinations (Bai et al., 2024), and AI alignment (Ji et al., 2023). Multi-step inference requires the models to follow instructions precisely to reach the correct conclusions. Models that adhere strictly to instructions provide clear intermediate reasoning steps, resulting in more transparent and interpretable outputs, which is essential for explainable AI. Strict procedural adherence reduces the risk of generating inaccurate or nonsensical information, thereby mitigating hallucinations by ensuring logical connections of distinct pieces of knowledge. Furthermore, ensuring that AI systems follow human instructions is fundamental to aligning their behavior with human intentions from the aspect of both safety and functionality.

Using ProcBench, we evaluated several state-of-the-art large language models (LLMs) to assess their instruction followability. Our evaluations of several state-of-the-art LLMs demonstrate a wide range of performance across tasks and complexity levels. Some models, such as **o1-preview** and **o1-mini**, performed consistently well on simpler tasks, accurately following multi-step instructions. However, as the complexity increased with longer sequences, even these models exhibited a significant drop in performance, highlighting their limitations in handling complex, multi-step reasoning. These findings emphasize the need for future improvements in procedural reasoning and offer a pathway for advancing LLMs in this area.

## 2 RELATED WORK

### 2.1 BENCHMARKS FOR LARGE LANGUAGE MODELS

Various benchmarks have been proposed to assess the capabilities of LLMs across diverse domains. Some benchmarks evaluate knowledge, reading comprehension, and general reasoning skills in fields such as science, medicine, and law (Hendrycks et al., 2020b; Dua et al., 2019; Lai et al., 2017). Others focus on problem-solving and code generation capabilities (Suzgun et al., 2022; Chen et al., 2021). Mathematical reasoning is assessed through specific benchmarks (Saxton et al., 2019; Cobbe et al., 2021; Shi et al., 2023), while some are designed to evaluate LLMs' performance in software operations, such as executing commands and web browsing (Xi et al., 2024; Liu et al., 2023; Ma et al., 2024). Instruction-following capabilities have also begun to receive attention through recent benchmarks (Zhou et al., 2023). Due to the rapid development of LLMs, such benchmarks that require implicit knowledge described above generally require frequent updates by adding more difficult tasks, or have short useful lifespans (Martínez-Plumed et al., 2021). Furthermore, because these benchmarks tend to focus on specific task-oriented skills that LLMs have been trained on, they are not fully adequate for assessing the models' general intellectual capabilities, which is also pointed out in Chollet (2019).

In contrast to existing benchmarks, ProcBench focuses on assessing procedural reasoning, an essential component of complex problem-solving that remains underexplored. By isolating procedural

follow-up from domain-specific knowledge, ProcBench reveals significant limitations in the ability of LLMs to strictly adhere to detailed, multi-step instructions. This makes the challenges of procedural reasoning explicit and provides a new perspective for evaluating and improving LLMs in domains that require precise, sequential operations.

## 2.2 INSTRUCTION FOLLOWING

Instruction following (Lou et al., 2024) has become an important research area, particularly in the context of LLMs (Zhou et al., 2023; Kim et al., 2024; Mishra et al., 2022). The primary goal in this field is to evaluate whether models can accurately interpret and execute given instructions. However, much of the existing research focuses on the final output, with less attention paid to the reasoning process that leads to that output.

Our research links Instruction following to reasoning, positioning it as a specialized form of multi-step reasoning. We emphasize the importance of evaluating not only whether the final output follows the instructions but also the intermediate steps taken during the problem-solving process, distinguishing our work from previous studies.

Remove letters in a given list from a given string step by step; at each step remove a single letter starting from the first element of the list.
Provide the final string along with the intermediate states after each step in the form of a list.
Do not include the initial state and final state in the list of intermediate states.

[Question]
String: hchouumkd
Steps: c, u, h, k, d, o, h, m

initial state: hchouumkd
step1: hhouumkd
step2: hhoumkd
step3: houmkd
step4: houmd
step5: houm
step6: hum
step7: um
final state: u

(a) An example of input prompt  (b) An example of ground truth label

Figure 1: An example from the task *DeleteChar*. (a) shows the input prompt, where the task is to iteratively remove specific letters from the given string according to the provided steps. (b) represents the ground truth label, which demonstrates the intermediate and final states of the string after performing each step of deletion.

## 3 PROCBENCH

In this section, we introduce ProcBench, a benchmark dataset for testing LLMs' instruction followability. The models are asked to solve simple but long step-by-step tasks by precisely following the provided instructions. Each step is a simple manipulation of either a string, a list of strings, or an integer number. There are 23 types of tasks in total, listed in Table 1. The tasks are designed to require only minimal implicit knowledge, such as a basic understanding of the English language and the ordering of alphabets. While the complexity increases with the number of steps, the tasks can essentially be solved by following the instructions without the need for specialized knowledge. While these tasks are easy for humans regardless of their lengths as long as we can execute each step, LLMs may fail as the number of steps becomes larger.

## 3.1 STRUCTURE

Each example is composed of the combination of a template and a question. Each task is associated with a fixed template, which contains the procedure for solving the question. A concrete example of this combination is shown in Figure 1a, along with the corresponding intermediate states and final state as the ground truth in Figure 1b. Additional templates can be found in Appendix A. The question represents the specific problem and is generated by the Generator. The Generator also produces the correct answer and the intermediate states leading to that answer simultaneously. Since the questions are generated by the Generator, it is easy to increase the number of examples in our dataset. However, for the convenience of evaluation, we provide a fixed dataset. We set the number

Table 1: List of tasks.

| Name | Type | Description |
|------|------|-------------|
| Compare | str / list[str] | Compare a target string with a list and find the first exact match. |
| Compose | list[str] | Replace adjacent characters based on rules. |
| Copy | str | Concatenate strings based on a list of indices. |
| Count | list[str] / int | Count alphabets and numeric characters, then compute a product. |
| Count2 | list[int] | Count alphabets word by word, ignoring case. |
| Cumulate | int | Add or multiply numbers based on operations. |
| Decode | str | Decode a compressed string representation. |
| Decompose | str | Replace characters based on rules while replacement is possible. |
| DeleteChar | str | Remove characters step by step from a string. |
| DeleteWord | str | Remove words step by step from a sentence. |
| Encode | list[str] | Encode a binary string by grouping identical characters. |
| FillWord | str | Replace numbers in a sentence with words from a list. |
| FindCyclic | list[str] / str | Find a letter by moving a specified number of steps cyclically. |
| Gather | str | Extract and concatenate substrings based on index sets. |
| MoveCyclic | str | Move a character cyclically in an array. |
| PushPop | str | Add or remove characters using push/pop actions. |
| Rhythm | str | Pair and concatenate numbers and characters cyclically. |
| Rotate | str | Rotate a substring within a string step by step. |
| Search | list[int] | Count substring occurrences in a list of strings. |
| Sort | str | Sort a string alphabetically by swapping characters. |
| Split1 | list[str] | Split a string at specified positions. |
| Split2 | list[str] | Split substrings based on index pairs. |
| Substitute | str | Replace characters in a string based on a list of pairs. |

of steps for each problem from 2 to 25, generating 10 examples per number. Therefore, each task consists of 240 examples, with a total of 5,520 examples. We further classify the step counts of 2 to 6 as Short, 7 to 16 as Medium, and 17 to 25 as Long, and aggregate metrics at these levels as well.

The models receive the combination of a template and a question and are required to provide not only the final state but also the intermediate states. Thus, the responses are more complex than simply providing words or choices. The elements contained in the intermediate and final states are of types int, str, and list. The list type contains int or str as its elements. The types of these elements differ by task, as listed in Table 1. The model's predictions must be converted into a JSON format that adheres to these types to facilitate evaluation through the metric functions.

## 3.2 METRICS

We introduce *Prefix Accuracy (PA)* as a metric for evaluating sequence-based tasks, with a primary focus on assessing how accurately a system adheres to a specific procedure to solve a question. These tasks often involve multi-step procedures or free-form answer generation, where even small deviations from the correct steps can lead to significant mismatches. The sequences in question represent complex state transitions, making precise adherence to the intended procedure critical for successful problem-solving.

Let $T = (t_0, t_1, t_2, \ldots, t_N)$ represent the target sequence of length $N + 1$, where $t_0$ corresponds to the initial state provided in the question and is not required to be predicted. The parameter $N$ denotes the number of steps necessary to solve the problem, which we refer to as the *Problem Length*. Similarly, let $P = (p_1, p_2, \ldots, p_M)$ denote the predicted sequence of length $M$.

We define the *Prefix Match Length (PML)* as the length of the longest contiguous prefix where the elements of the predicted sequence match those of the target sequence exactly. Formally, let $k$ be the largest index such that $t_i = p_i$ for all $1 \leq i \leq k$. Then,

$$\text{PML}(T, P) = \max \{k \mid t_i = p_i \text{ for all } 1 \leq i \leq k\}.$$

Here, the comparison of sequence elements $t_i = p_j$ denotes an exact match regardless of type. Specifically, for each of `int`, `str`, and `list`, it represents complete equality as integers, characters, or lists, respectively. In the case of lists, a match is considered to be 1 only if all corresponding elements in the lists are identical.

Using the PML, we define *Prefix Accuracy (PA)* as the ratio of the longest matching prefix to the length of the longer sequence between the target and predicted sequences. This normalization ensures that the metric is bounded between 0 and 1, where 1 indicates a perfect match. Formally,

$$\mathrm{PA}(T, P) = \frac{\mathrm{PML}(T, P)}{\max(N, M)}.$$

PA penalizes both over- and under-prediction by considering the length of the longer sequence, thereby reducing the score for deviations in either direction.

In addition, we introduce *Sequential Match (SM)* as a binary indicator of whether the predicted sequence perfectly matches the target sequence from the first element to the last. SM is defined as:

$$\mathrm{SM}(T, P) = \begin{cases} 1 & \text{if } \mathrm{PA}(T, P) = 1, \\ 0 & \text{otherwise.} \end{cases}$$

SM captures cases where the predicted sequence adheres fully to the target sequence across all steps, making it a stringent indicator of complete procedural correctness.

Finally, we introduce *Final Match (FM)* as a binary indicator of whether the final elements of the two sequences match. This metric captures whether the final state, or solution, predicted by the model aligns with the target outcome, regardless of intermediate discrepancies. Formally,

$$\mathrm{FM}(T, P) = \begin{cases} 1 & \text{if } t_N = p_M, \\ 0 & \text{otherwise.} \end{cases}$$

FM complements PA by ensuring that final solutions are evaluated independently of intermediate state matches.

## 4 EXPERIMENT

### 4.1 EXPERIMENTAL SETUP

We evaluate the performance of seven state-of-the-art models using our benchmark, which covers a variety of task types and complexities. The models selected for evaluation include Claude-3.5-sonnet (Anthropic, 2024), Mistral-large (Jiang et al., 2023), Gemini-1.5-Pro (Google, 2024), GPT-4o, GPT-4o-mini (OpenAI, 2023), o1-mini, and o1-preview (OpenAI, 2024a).

The tasks presented to the models require generating sequences rather than simple question-and-answer pairs. Given that the output of LLMs is generally provided in free-form text, we convert the responses into a structured JSON format (OpenAI, 2024b) to facilitate evaluation (See Appendix C for details). This transformation process, performed by GPT-4o, is uniformly applied to all models, and the evaluation metrics defined in Section 3.2 are computed based on this standardized format. It is important to note that the results reflect not only the models' raw accuracy but also the impact of the conversion process on the final evaluation scores.

### 4.2 RESULTS

**Model Performance in Summary.** Table 2 provides a comprehensive comparison of model performances across varying task difficulty levels—Short, Medium, and Long—evaluated through the metrics of PA (Prefix Accuracy) and SM (Sequential Match). The **o1-preview** model consistently leads across most categories, particularly excelling in the Medium and Long tasks, where it achieves the highest scores for both PA and SM. In contrast, **o1-mini** demonstrates a competitive edge in simpler tasks, outperforming o1-preview in the Short task with a PA of 0.801 and SM of 0.722.

**Performance Across Problem Length and Stepwise Predictions.** To further analyze how model performance is affected by problem length $N$, Figure 3 displays the results across four key metrics:

Table 2: Model Performance Across Different Difficulty Levels. **Prefix Accuracy (PA)** measures the ratio of the longest matching prefix between the predicted and target sequences, normalized by the length of the longer sequence.**Sequential Match (SM)** is a binary metric indicating whether the predicted sequence exactly matches the target sequence from start to finish. The full definitions of the metrics can be found in Section 3.2.

| Model | Short | | Medium | | Long | | Overall | |
|---|---|---|---|---|---|---|---|---|
| | PA | SM | PA | SM | PA | SM | PA | SM |
| Claude-3.5-Sonnet | 0.589 | 0.455 | 0.382 | 0.221 | 0.255 | 0.115 | 0.378 | 0.230 |
| Mistral-Large | 0.495 | 0.366 | 0.384 | 0.239 | 0.265 | 0.142 | 0.362 | 0.229 |
| Gemini-1.5-Pro | 0.329 | 0.167 | 0.230 | 0.117 | 0.159 | 0.071 | 0.224 | 0.110 |
| GPT-4o | 0.631 | 0.396 | 0.437 | 0.285 | 0.344 | 0.204 | 0.443 | 0.278 |
| GPT-4o-mini | 0.408 | 0.201 | 0.220 | 0.089 | 0.143 | 0.039 | 0.230 | 0.093 |
| o1-mini | **0.801** | **0.722** | 0.681 | 0.484 | 0.508 | 0.214 | 0.641 | 0.432 |
| o1-preview | 0.799 | 0.656 | **0.736** | **0.563** | **0.599** | **0.333** | **0.698** | **0.496** |

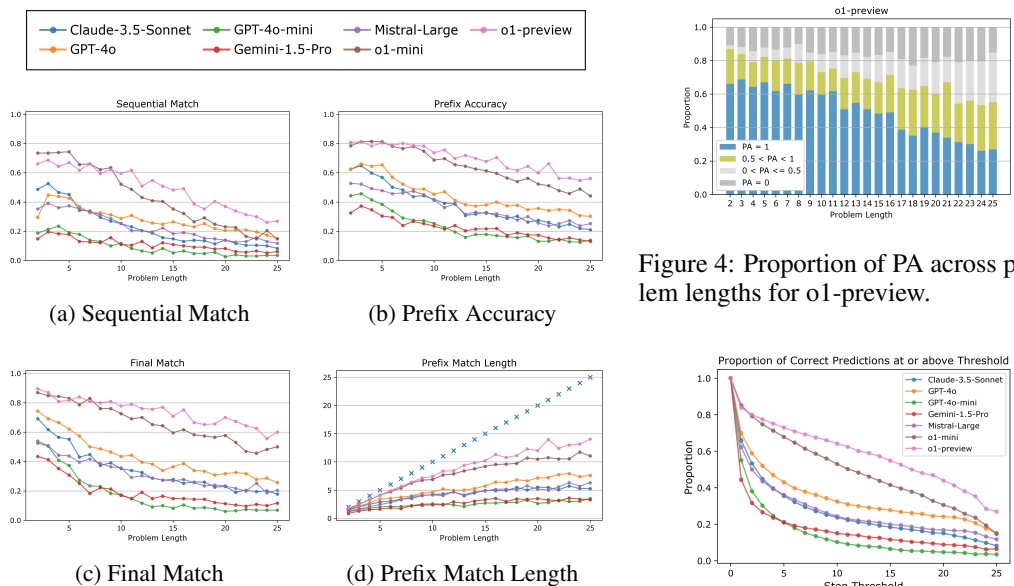

(a) Sequential Match

(b) Prefix Accuracy

Figure 4: Proportion of PA across problem lengths for o1-preview.

(c) Final Match

(d) Prefix Match Length

Figure 3: Performance Metrics: SM, PA, FM, and PML across models and problem length.

Figure 5: Proportion of Correct Predictions at or above step threshold.

SM, PA, FM, and PML. These metrics provide a detailed view of the models' effectiveness in solving sequence-based tasks as $N$ increases.

By definition, SM is the most strict of the metrics. Indeed, as shown in Figure 2a, SM exhibits the most pronounced decline as the Problem Length increases. While PA shows a similar trend to SM, its decline is more gradual (Figure 2b). Additionally, PA follows a pattern similar to PML (Figure 2d), but its normalization allows for model comparisons independent of problem length. FM was originally defined based on the inherent difficulty of our proposed tasks and the common assumption that only the final answer needs to be correct. However, as observed in the averaged visualizations in Figure 2c, this metric behaves almost identically to SM and PA in practice. Figure 2d focuses on the PML metric, which reveals that the average PML increases with problem length but plateaus after a certain point. This suggests that models have inherent limitations in the number of inference steps they can reliably manage.

Figure 4 illustrates the distribution of PA across different problem length bins for **o1-preview**, the model with the highest overall performance. The visualization reveals that the proportion of errors at the initial step remains nearly constant, regardless of problem length $N$. While the model demon-

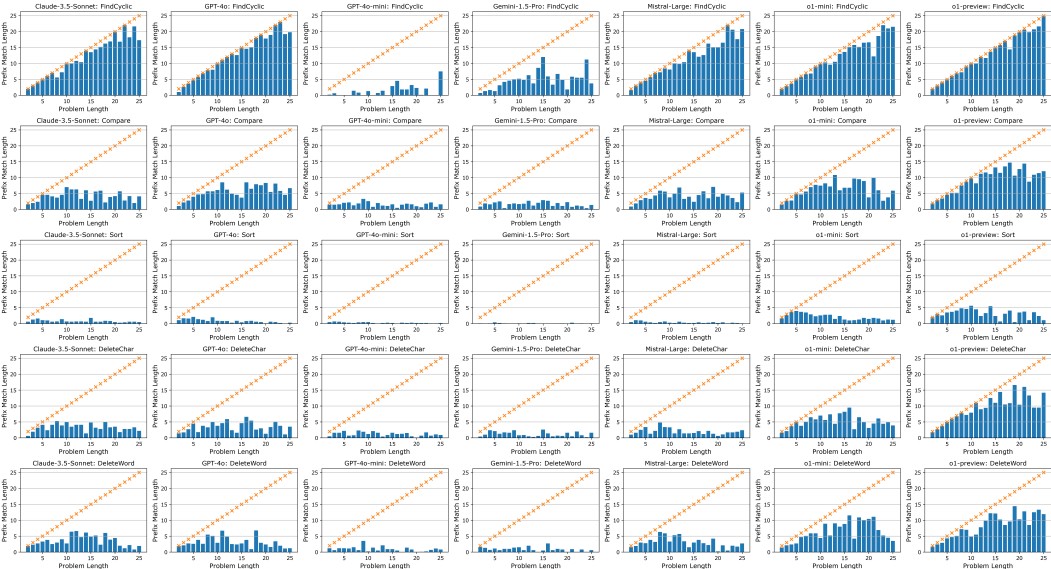

Figure 6: Prefix Match Length (PML) for different problem lengths across all models and three tasks; FindCyclic, Compare, Sort, DeleteChar and DeleteWord. Each bar in the graph represents the average PML for a given problem length, with separate graphs for each model-task pair.

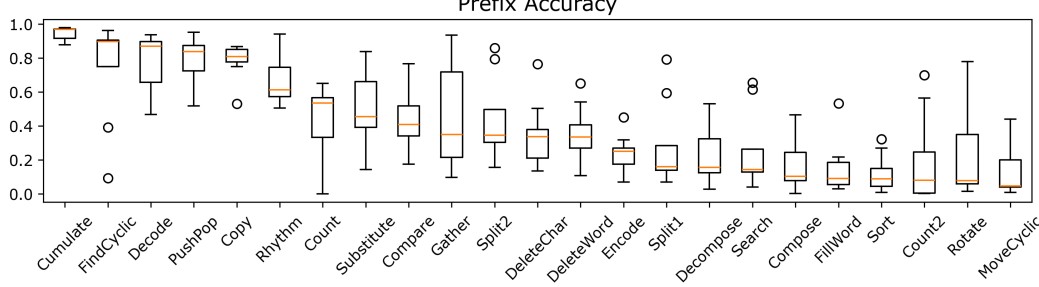

Figure 7: Model performance across all 23 tasks. The metrics shown are Prefix Accuracy. Tasks are ordered by the median PA.

strates high accuracy for shorter questions, its performance gradually declines as $N$ increases, with longer questions resulting in lower PA. A similar pattern can be observed in other models, as shown in Figure 12 in the Appendix.

Figure 5 visualizes the proportion of correct predictions made by each model at or above a given step threshold. Each point represents the proportion of questions for which the model successfully predicted beyond a particular step threshold $N$. Similar to an ROC curve, models with curves that remain higher and shift to the right demonstrate stronger performance on tasks with longer sequences. **o1-preview** and **o1-mini** exhibit superior performance, with their curves declining more gradually, indicating their ability to handle longer step sequences effectively. In contrast, other models experience a sharp drop in accuracy around the 5-step mark, reflecting their limited capacity to maintain correct predictions over extended sequences.

**Impact of Tokenization on Task Performance** To examine the impact of tokenization methods on task accuracy, we analyzed two tasks, DeleteChar and DeleteWord, which differ in the granularity of string manipulations. The DeleteChar task involves removing characters from random strings, requiring fine-grained token manipulations. In contrast, DeleteWord entails removing words from sentences in the Wikipedia English corpus, reflecting token-level operations where frequent words are often single tokens and less common ones span multiple tokens.

Contrary to our hypothesis that DeleteChar would be more challenging due to its reliance on tokenization-related transformations, results (Figure 6) showed no significant difference in accuracy between the two tasks, with DeleteChar even slightly outperforming DeleteWord. This suggests that tokenization may have a limited impact on task performance, with other factors, such as task-specific properties or the model's internal representations, playing a larger role.

**Task-Specific Model Performance** To further assess model accuracy on specific tasks, Figure 6 illustrates PMLs across tasks such as FindCyclic, Compare, and Sort. The results show that some models retain high accuracy throughout all problem lengths in FindCyclic, while they show significant accuracy declines as the number of steps increases in Compare. Additionally, certain tasks like Sort consistently exhibit smaller PML values across steps, indicating their difficulties. All graphs can be found in Figure 13 and 14 in Appendix.

Lastly, Figure 7 presents the accuracy variation across the seven models for each of the 23 tasks in the dataset. Notably, tasks such as FillWord and Sort have been identified as particularly challenging for many models, with certain questions in these tasks frequently resulting in lower PA. If a substantial portion of questions within a task consistently result in PA = 0, this could indicate not merely high difficulty but also potential flaws in task design, such as internal contradictions or settings that render the problem unsolvable. Nevertheless, only 91 out of the 5,520 examples across the entire dataset exhibit a PA of 0 across all models, suggesting that the dataset is well-calibrated and that such issues of unsolvability or internal contradictions are minimal.

**Analysis of Prompt Strategies and Error Tendencies** We conducted additional experiments to analyze the impact of different prompt strategies on ProcBench tasks using four OpenAI models. The experiments, though limited in scope, evaluated half of the dataset by sampling 5 out of 10 questions for each problem length and yielded meaningful insights.

The few-shot setting, where three examples (including Question, Intermediate States, and Final State) were inserted between the Template and the Question, improved accuracy across many tasks. This suggests that concrete examples effectively reduce ambiguity and support complex outputs. In contrast, the one-go setting, which omits intermediate states and outputs only the final state, resulted in lower accuracy. The higher performance in the standard ProcBench setting underscores the importance of intermediate states, which appear to function similarly to Chain-of-Thought reasoning, aiding step-by-step inference. See Figure 16 and 17 in Appendix D in details.

**Error Analysis of Tasks with PA=0** An error analysis revealed that the number of instances with PA=0, where none of the models provided correct solutions, varied significantly across tasks. Each task contains a total of 240 instances, with 10 questions per step from steps 2 to 25. The most prominent errors occurred in the FillWord and MoveCyclic tasks. For example, in the case of the FillWord task, 40 out of 240 instances (16.6%) were unsolved by any model, even in the worst case. Considering that the template is always fixed for each task, this suggests that the errors are likely attributable to the high difficulty of certain tasks.

A closer examination of these PA=0 cases revealed distinct patterns of errors for each task. For MoveCyclic, while the specific tendencies of incorrect predictions varied across models, a significant proportion of mistakes were related to format transformation errors. In contrast, for FillWord, errors were often due to models incorrectly determining the order in which blanks should be filled. These findings highlight the need for models to better address task-specific challenges and systematic format transformations.

## 5 DISCUSSION

**Relationship Between Instruction Following and Reasoning.** The relationship between Instruction following and reasoning is a fascinating one. For humans, the most challenging aspects of reasoning often involve the application of knowledge, particularly implicit knowledge. In contrast, simply following instructions is generally not considered reasoning. However, we propose that Instruction following can be understood as a specialized form of reasoning, particularly when it is disentangled from implicit knowledge and focuses on scenarios where the path to the goal is explicitly defined. Although this may not initially seem like reasoning, once one successfully navigates the search for the correct procedure and applies the relevant knowledge, it becomes closely aligned with the types of problems we seek to address. In this sense, our approach deconstructs the reason-

ing process. While the ultimate reasoning systems may not explicitly separate these functions, they should still be capable of solving the kinds of problems we address.

We developed a dataset to explore this connection, and our results show that models recognized for their strong reasoning capabilities, such as o1-preview and o1-mini, performed well. This suggests a qualitative link between reasoning ability and the capacity to follow instructions. However, the experiments also revealed that even tasks that may seem straightforward or merely tedious to humans—tasks that appear self-evident—are not consistently solved by the models. On the other hand, these models demonstrate strong reasoning capabilities in domains like law or physics (OpenAI, 2023; Google, 2024). Since the effective application of knowledge can reduce the number of computational steps, it suggests that state-of-the-art LLMs may be more adept at leveraging knowledge to solve complex problems rather than excelling at multi-step procedural reasoning itself. This underscores a fundamental challenge in the current deep learning paradigm, where many models struggle with intricate reasoning tasks unless they can heavily rely on prior knowledge.

**Minimal Implicit Knowledge.** While the ideal goal is to eliminate all implicit knowledge requirements, some minimal assumptions are inevitable. For example, we assume a basic understanding of the English language, the order of the alphabet, and that numbers such as 0, 1, 2, and so on represent numerical values. However, these assumptions are deliberately kept minimal and are significantly less specialized compared to the knowledge required for tasks in fields like physics, chemistry, law, or mathematics. By focusing on such foundational concepts, the dataset preserves a structured challenge that emphasizes reasoning and procedural execution rather than relying on domain-specific knowledge.

**Expected Use Cases and Limitations.** The simplest and most straightforward use of our dataset is for evaluating LLMs, particularly with respect to their reasoning capabilities. This is the primary intended application, allowing researchers to assess how well a new model handles multi-step reasoning tasks.

Additionally, ProcBench can be used to evaluate variations of methods such as In-Context Learning or Chain-of-Thought reasoning (Wei et al., 2023). However, we do not envision the use of task-specific prompts for each of the 23 distinct tasks in the dataset, as this would introduce domain-specific knowledge. Such prompts might enable the model to skip significant portions of the actual reasoning process, which would defeat the purpose of evaluating its raw reasoning capabilities. An extreme example of this would be programming-based solutions, where directly introducing task-specific solvers should be avoided. If a general-purpose model, such as one with coding capabilities, can solve tasks without specific tuning, this reflects its versatility. However, in such cases, the intended measurement of multi-step Instruction followability would no longer be feasible within this dataset.

Although the primary focus is on the current LLM paradigm, ProcBench is still applicable to traditional machine learning models, such as those used in inductive programming, which learn from concrete examples. However, the fixed dataset provided for evaluation is unlikely to be sufficient for training such models. In this case, the Generator could be utilized to augment the dataset, enabling a model to be constructed from scratch with the goal of following procedural instructions.

## 6 CONCLUSION

We introduced ProcBench, a benchmark designed to assess LLMs on their ability to follow explicit, multi-step instructions. By concentrating on tasks that require minimal implicit knowledge, ProcBench allows us to evaluate the procedural reasoning capabilities of models independent of their reliance on domain-specific knowledge. Our results show that while state-of-the-art models like o1-preview and o1-mini perform well on tasks involving shorter steps, they face significant difficulties as the step length increases. This highlights a critical limitation in current LLMs: despite excelling in knowledge-driven tasks, they struggle to consistently follow detailed procedural instructions when faced with more complex, multi-step reasoning.

Our findings emphasize the distinction between knowledge-based reasoning and instruction following, an area where LLMs have yet to achieve consistent mastery. Enhancing the ability of these models to precisely follow instructions will be key to improving their performance in more complex problem-solving scenarios. Future work will expand ProcBench to encompass a broader range of

tasks and further investigate how explicit instruction-following capabilities can be more effectively integrated into models trained on traditional benchmarks. This will contribute to developing systems that can reliably handle multi-step reasoning across diverse domains.

## REPRODUCIBILITY

Details regarding the construction of the benchmark and the models used in the experiments can be found in Section 3, Section 4.1 and Table 3. The dataset we created, the code used to generate it, the prediction results, and the evaluation results will be made publicly available after the paper is accepted for publication.

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

# A DATASET EXAMPLES

## A.1 EXAMPLE TEMPLATES

Here, we provide detailed examples of 3 out of the 23 templates used in our experiments. The remaining templates can be found in the dataset that has been made publicly available. Please refer to the dataset for the full set of templates.

---

**Template of Sort**

Sort a given string into alphabetical order step by step by swapping two characters. Starting with the first alphabet in alphabetical order 'a' and the first position of the string, repeat the following process until the end of the alphabetical order 'z'. At each step, search for the alphabet in the string from left to right. If you find it, swap it with the character at the current position. Then, move to the next position of the string. If the alphabet is not found, do nothing. Repeat the step until the whole string has been searched, and move on to the next alphabet.

Provide the final sorted string along with the intermediate strings after each swap in a list ignoring the steps with no change. Do not include the initial state and final state in the list of intermediate states.

---

**Template of Rotate**

Rotate a substring within a string by partitioning characters from original string's index m to n - 1, where the indices m and n are given in the form of (m, n) in 0-based indexing. For each index pair, shift every character within the partitioned substring to the right by one, and wrap the right-most character around to the beginning of the substring. Provide the final string along with the intermediate states after each step in a list. Do not include the initial state and final state in the list of intermediate states.

---

**Template of MoveCyclic**

Move "x" in a given array consisting of "x" and "-" following a list of operations, where the operation is given as a pair of the direction, left or right, and the amount of movement. If "x" reaches the beginning of the array, move it to the end, and vice versa.

Provide the final state of the array and the intermediate states after each movement in a list. Do not include the initial state and final state in the list of intermediate states.

---

Figure 8: Templates for the tasks of Sort, Rotate, and MoveCyclic.

## A.2 EXAMPLE PROMPTS

---

**A prompt of Substitute**

Replace characters in a given string according to a given list of character pairs step by step as follows.
Start from the first character in the string.
At each step, if the current target character matches the first element of any pairs in the list, replace the character with the second element of the pair. Then, move on to the next character in the string.
Repeat the step until the end of the string.
Provide the final string along with the intermediate strings after each step in a list.
Do not include the initial state and final state in the list of intermediate states.
[Question]
Pairs:
(z, r)
(2, v)
String:
2z

---

**A prompt of Rhythm**

Given two sequences, where one is composed of numbers and the other is of characters, form pairs of a number and a character extracted from the sequences, respectively, in the form of a string.
Then, follow the procedure below.
Initialize an empty string and start from the first element of each sequence.
At each step, append the number and then the character obtained from the sequences to the string, and move to the next elements in both sequences.
If the end of either sequence is reached, wrap around to the beginning.
Repeat the process for N steps.
Provide the final string along with all the intermediate strings in a list.
Do not include the initial state and final state in the list of intermediate states.
[Question]
Sequence 1: 8, 6, 8, 7
Sequence 2: a, a, a, b, a, b, a, b
N: 5

---

**A prompt of Encode**

Encode a given string composed of '0's and '1's by following the procedure below.
Starting from the beginning of the string, count the number of the same character in series, and record the result in the form of "the character"_"the number of the character" at each step.
Repeat this step until the end of the string.
Provide the final result string along with the intermediate states in a 2D array, where each row has the list of the encoded results after each step.
Do not include the initial state and final state in the list of intermediate states.
[Question]
String:
0000000111111111000000011

---

Figure 9: Prompts from the tasks of Substitute, Rhythm and Encode.

## B   BASELINE MODELS

Table 3: Information of baseline models

| Name | Version | URL |
|------|---------|-----|
| GPT-4o | gpt-4o-2024-08-06 | `https://cdn.openai.com/gpt-4o-system-card.pdf` |
| GPT-4o-mini | gpt-4o-mini-2024-07-18 | `https://cdn.openai.com/gpt-4o-system-card.pdf` |
| o1-preview | o1-preview-2024-09-12 | `https://cdn.openai.com/o1-system-card-20240917.pdf` |
| o1-mini | o1-mini-2024-09-12 | `https://cdn.openai.com/o1-system-card-20240917.pdf` |
| Gemini-1.5-Pro | gemini-1.5-pro-latest | `https://deepmind.google/technologies/gemini/pro/` |
| Claude-3.5-sonnet | claude-3-5-sonnet-20240620 | `https://ai.meta.com/blog/llama-3-2-connect-2024-vision-edge-mobile-devices/` |
| Mistral-large | mistral-large-2407 | `https://mistral.ai/news/mistral-large-2407/` |

## C   STRUCTURED OUTPUTS

The transformation into JSON was performed using the Structured Outputs service provided by OpenAI, utilizing GPT-4o as the model. The Python API facilitates the conversion of text into JSON that adheres to a specified schema. As shown in Table 1, while the intermediate and final states of each task differ in type, these states can be extracted from the model's free-form responses by defining appropriate classes for each type. The system prompt used is provided in Figure 10, where the model's predicted text is inserted accordingly.

It is worth noting that the tasks in ProcBench inherently include the formatting of the model's output as part of the task itself. Consequently, this step should be regarded as a component of the model rather than as part of the evaluation function.

---

**System Prompt for Formatting**

You are an expert at structured data extraction. You will be given unstructured text and should convert it into the given structure. The task consists of a problem statement and a person's answer. The answer is in free-form text, but it needs to be formatted for evaluation purposes. Please convert the free-form text into the following JSON format. Do not include the final state in the last element of the intermediate list.

Figure 10: System prompt used by GPT-4o for structured output formatting.

# D    ADDITIONAL RESULTS

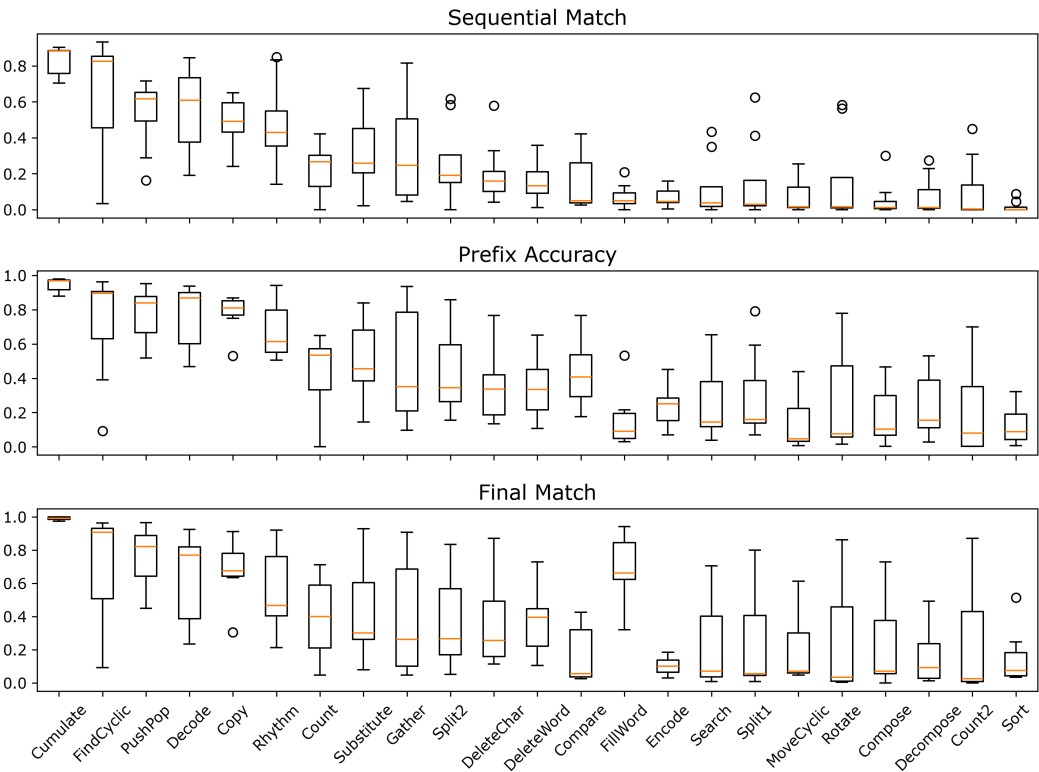

Figure 11: Model performance across all 23 tasks. The metrics shown are Sequential Match (SM) (top), Prefix Accuracy (PA) (middle), and Final Match (FM) (bottom). Tasks are ordered by the median SM score.

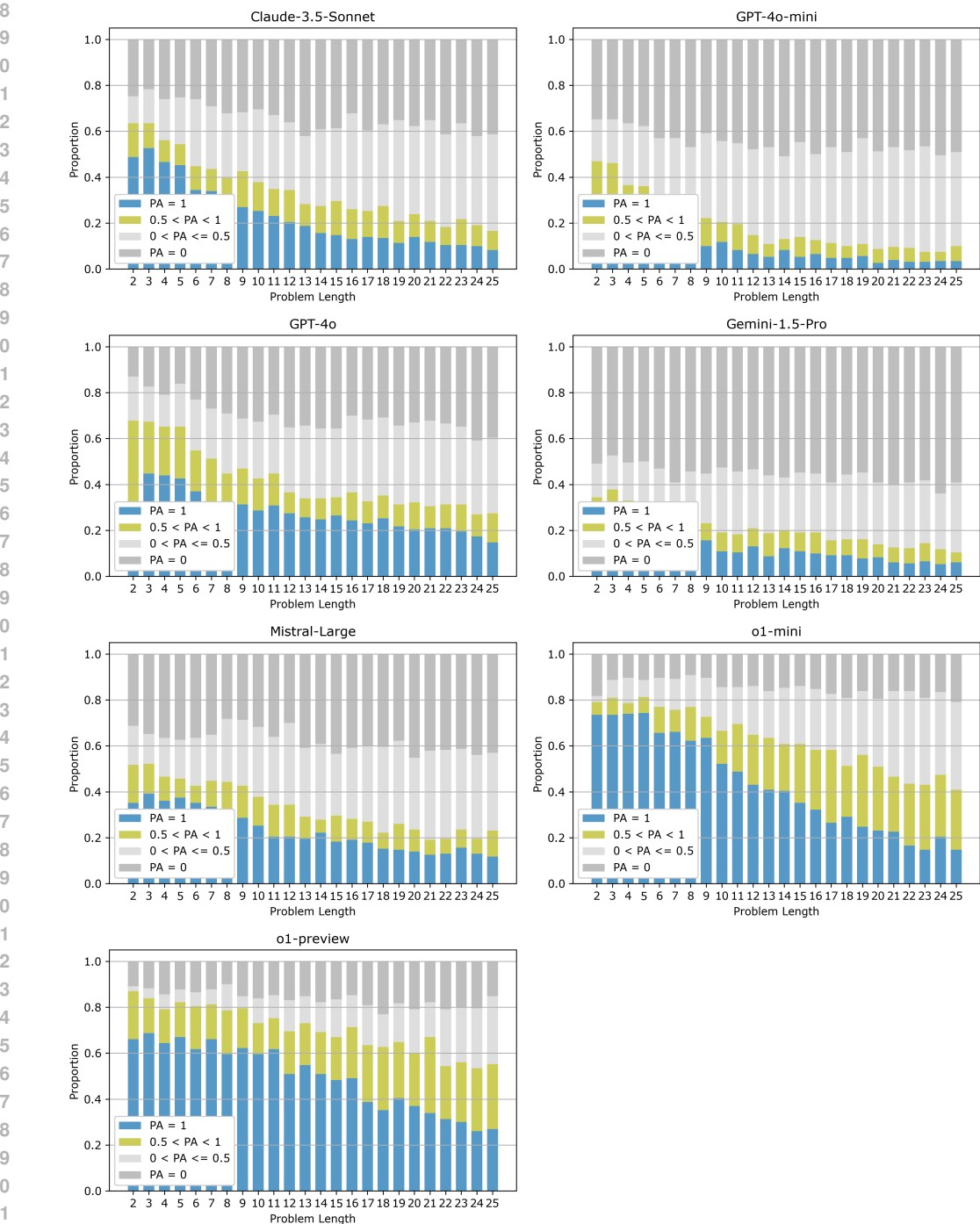

Figure 12: Proportion of PA across problem lengths for all models.

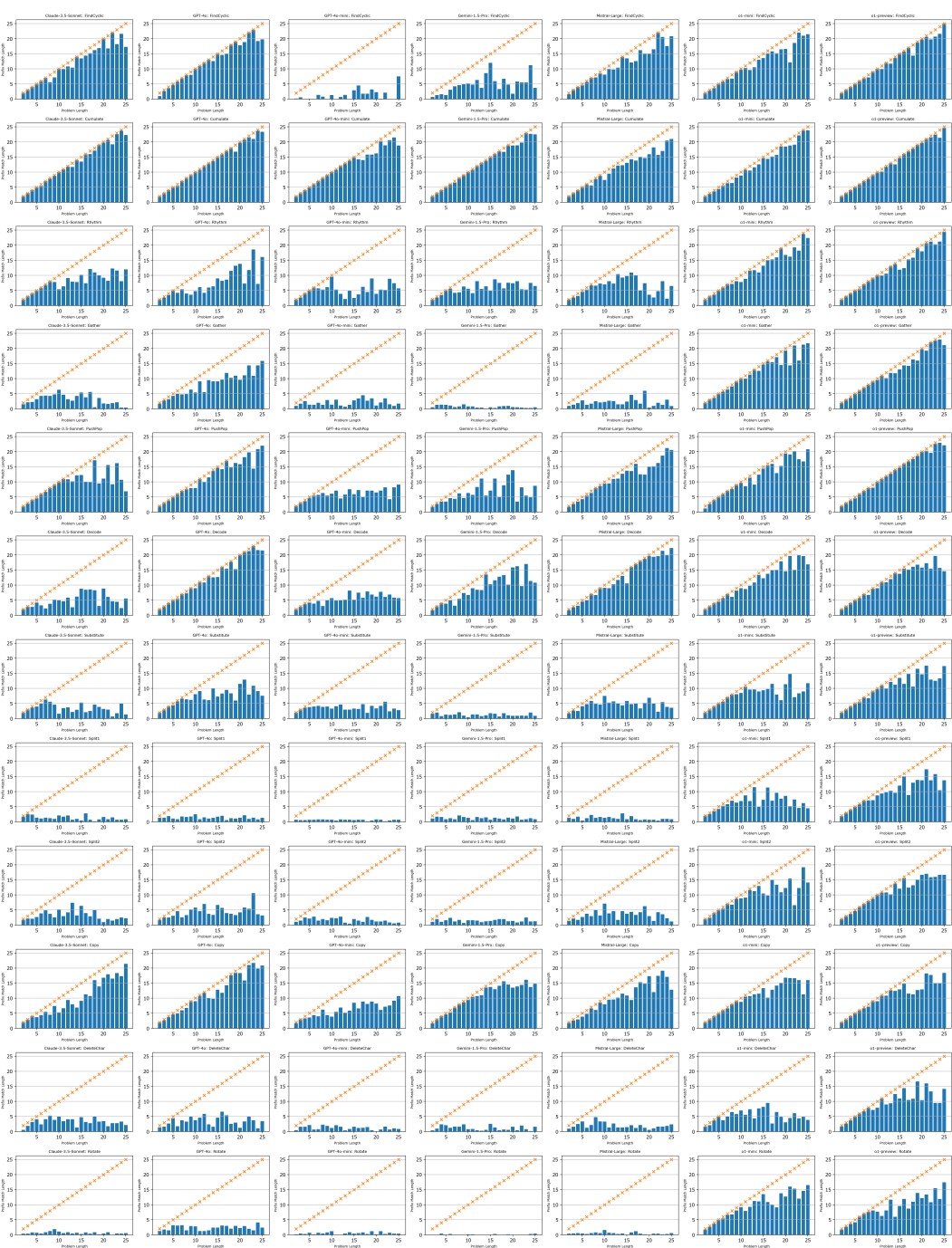

Figure 13: Prefix Match Length (PML) for different problem lengths across all models and tasks. Each bar in the graph represents the average PML for a given problem length, with separate graphs for each model-task pair. FindCyclic, Cumulate, Rhythm, Gather, PushPop, Decode, Substitute, Split1, Split2, Copy, DeleteChar and Rotate are shown.

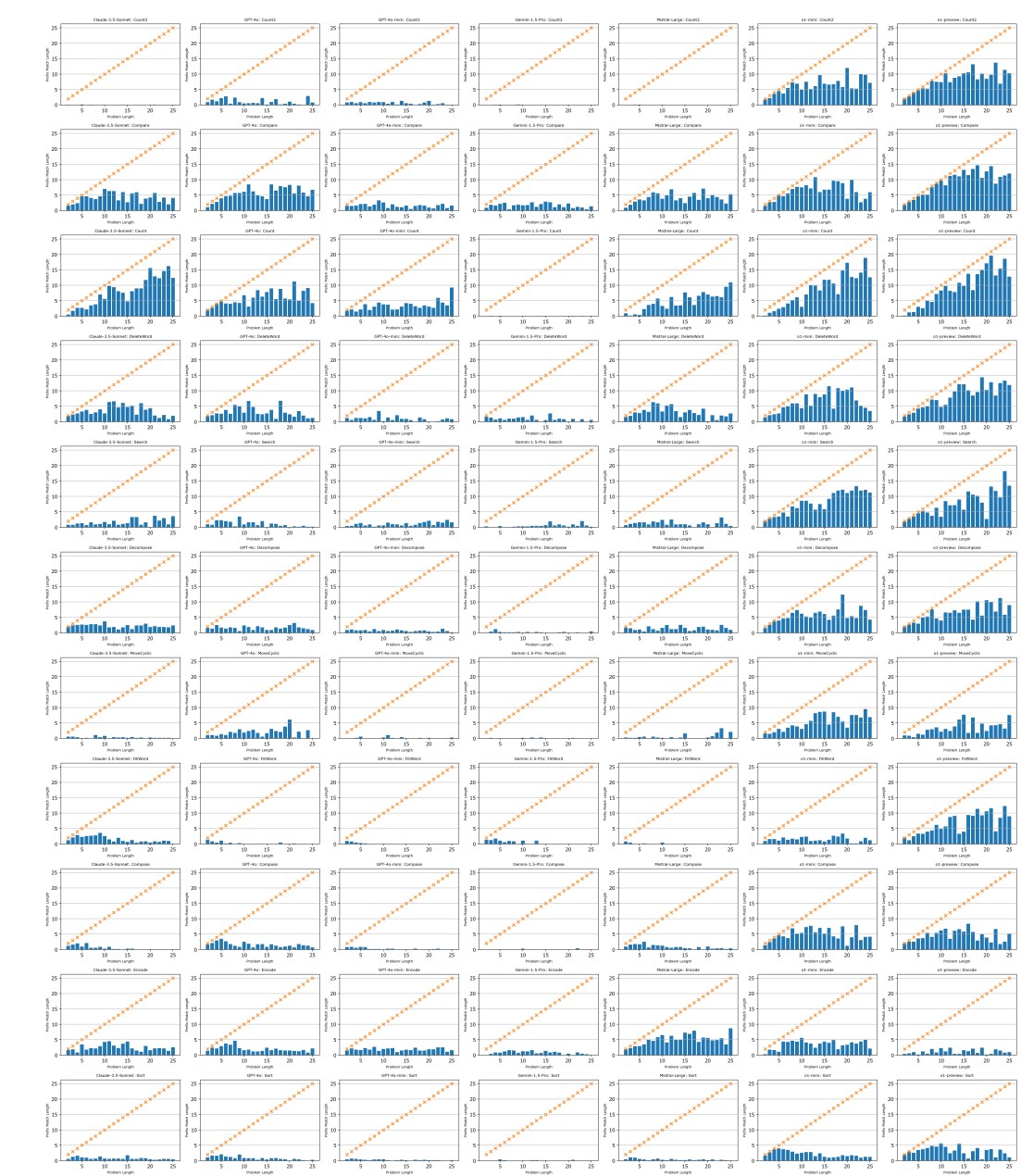

Figure 14: Prefix Match Length (PML) for different problem lengths across all models and tasks. Each bar in the graph represents the average PML for a given problem length, with separate graphs for each model-task pair. Count2, Compare, Count, DeleteWord, Search, Decompose, MoveCyclic, FillWord, Compose, Encode and Sort are shown.

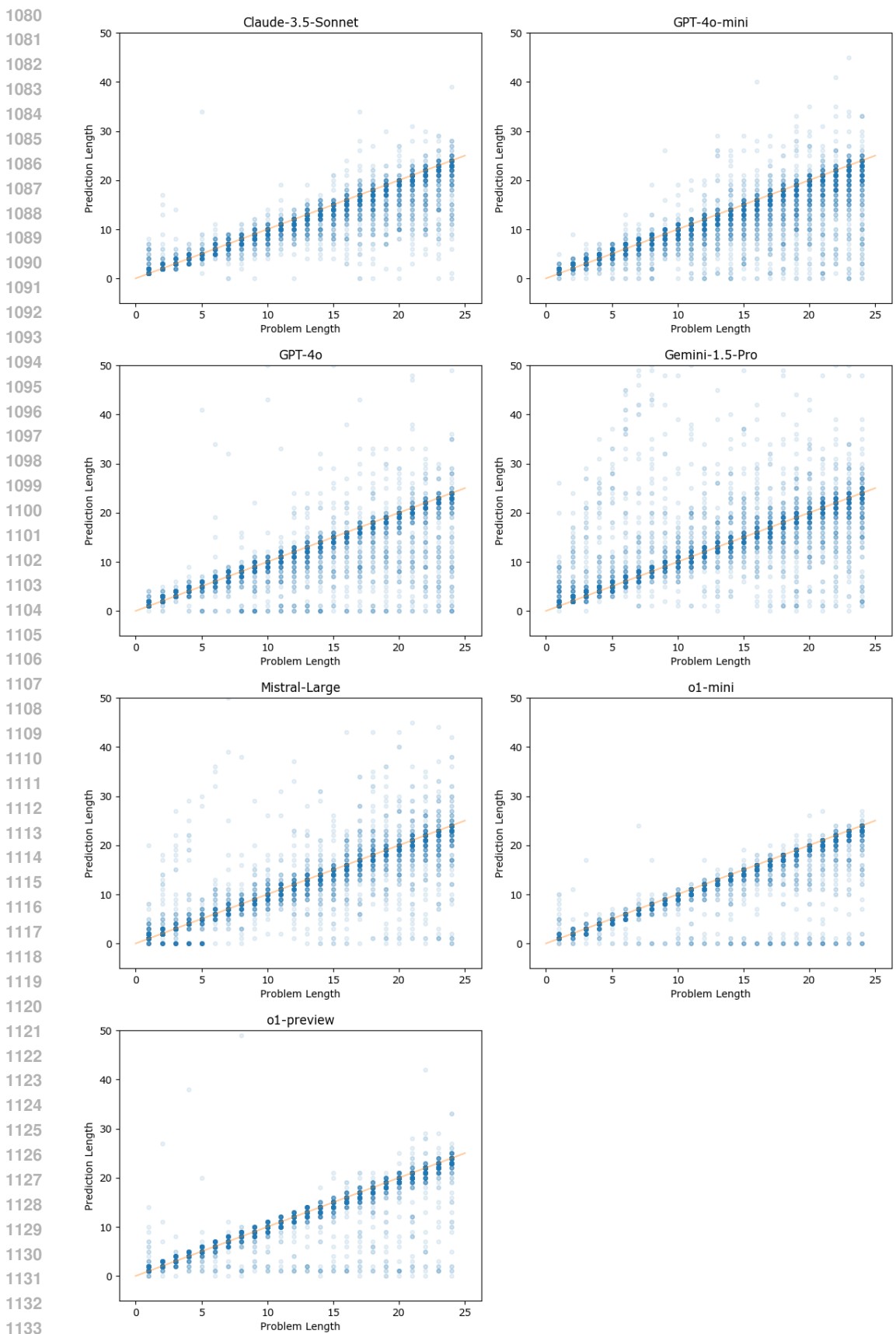

Figure 15: Problem Length vs Prediction Length. Gemini-1.5-Pro and Mistral-Large tend to give long predictions.

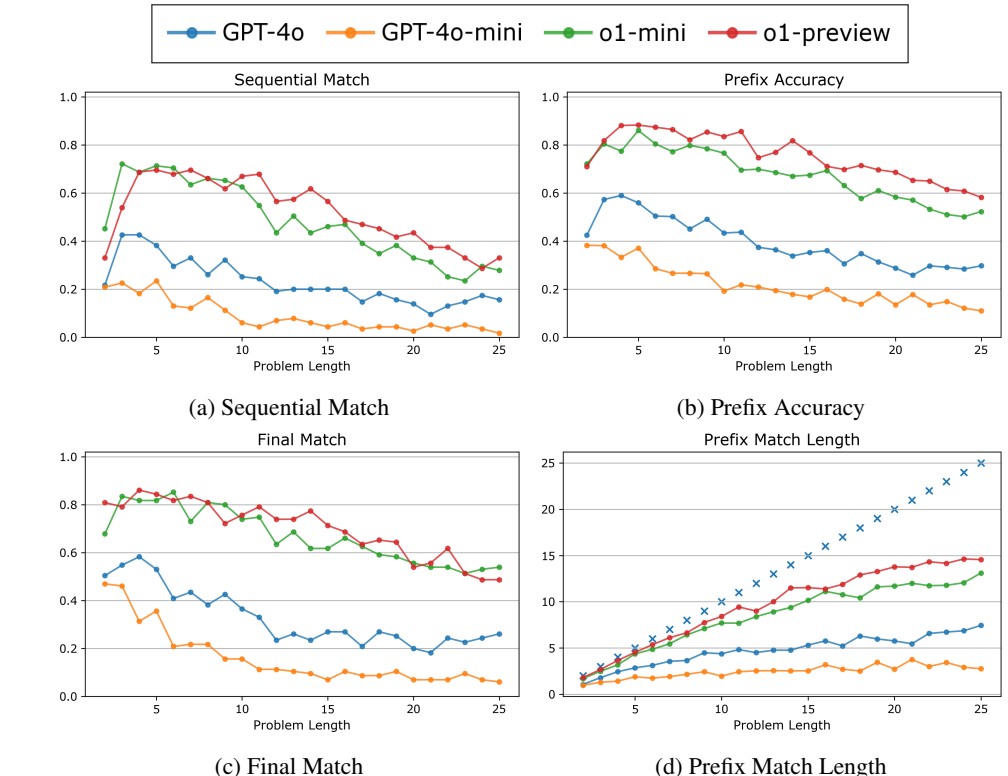

Figure 16: Performance Metrics: SM, PA, FM, and PML across models and problem length for few-shot setting.

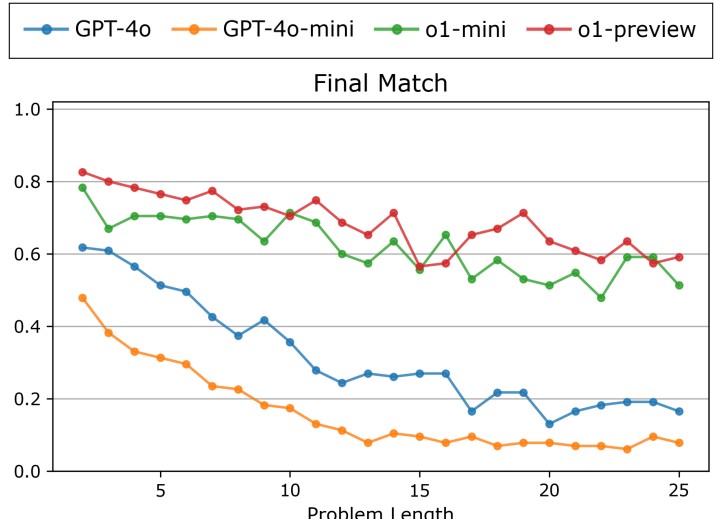

Figure 17: Performance Metrics: FM across models and problem length for one-go setting.

