# OpenReview forum: "ProcBench: Benchmark for Multi-Step Reasoning and Following Procedure"
_ICLR.cc/2025/Conference — Submitted to ICLR 2025_

### Official Review · Reviewer_PfvB · 2024-10-17

**Soundness:** 3
**Presentation:** 3
**Contribution:** 2
**Rating:** 3
**Confidence:** 4

**Summary:**

In this paper, the authors interpret the reasoning process as the exploration of paths, the utilization of knowledge, and the following of multi-step instructions. They propose ProcBench, a benchmark consisting of 23 subtasks designed to evaluate the multi-step instruction-following capabilities of large language models (LLMs), aiming to assess the reasoning ability of LLMs from a specific perspective. The authors evaluated the performance of seven models on ProcBench and conducted corresponding analyses.

**Strengths:**

- The authors provided a reasonable breakdown of reasoning abilities and, under controlled variables, introduced ProcBench to focus on exploring the multi-step instruction-following capabilities of models.
- The authors proposed metrics such as PML, PA, SM, and FM to assist in analyzing the performance of LLMs on ProcBench.
- The authors presented some interesting insights, including that the o1 series models exhibit a more gradual decline in performance as task difficulty increases, reflecting a fundamental difference between this series and other LLMs.

**Weaknesses:**

- When models engage in reasoning, they often do not follow a paradigm where they first plan all procedures and then execute them step by step, as typically reflected in benchmarks. Instead, they undergo a process of continuous exploration, following single-step instructions along the way. Therefore, the authors need to further verify the relationship between the multi-step instruction-following ability assessed by the benchmark and the actual reasoning ability. The experiments in the paper focus on ProcBench, but I suggest that the authors analyze the correlation between multi-step instruction-following ability and reasoning ability by statistically examining the reasoning performance of LLMs on relevant benchmarks and relating it to their performance on ProcBench.
- As the authors mentioned, 91 samples exhibit a PA of 0 across all models, which "could indicate not merely high difficulty but also potential flaws in task design." The authors are advised to conduct a case check to ensure the high quality of the benchmark.
- The authors provided quantitative statistics of LLM performance on ProcBench, but there is a lack of attribution analysis for model failures (as well as case studies), which is crucial for guiding model improvement.

**Questions:**

See Weaknesses



**After Rebuttal**

**Thanks for the author's response, although it was somewhat late. However, I noticed two points:**

1. The authors acknowledge that it is currently not possible to effectively establish the relationship between multi-step instruction-following ability and reasoning ability ("for future research"). However, this makes the mention of "MULTI-STEP REASONING" in the manuscript, particularly in the title, quite confusing. Since the relationship between ProcBench and reasoning ability cannot be validated, it raises the question of why reasoning ability is emphasized. I suggest the authors revise such phrasing, as it lacks rigor.

2. The authors have not responded to my third concern.

**Given these considerations, I believe the manuscript requires further refinement to meet the expected quality. Therefore, I have lowered my score.**

---

> ### Author Response · Authors · 2024-12-03
> **Response to Reviewer PfvB**
>
> >  the authors need to further verify the relationship between the multi-step instruction-following ability assessed by the benchmark and the actual reasoning ability.
>
> Understanding the relationship between multi-step instruction-following ability and reasoning ability is indeed an important direction for future research. However, our current work focuses on introducing ProcBench and validating its utility as a benchmark for evaluating multi-step instruction-following ability. While correlating this ability with reasoning performance across other benchmarks is valuable, it falls outside the scope of this paper. We hope to address this in future work to include cross-benchmark analysis and deeper investigations into the interplay between instruction following and reasoning capabilities.
>
> > As the authors mentioned, 91 samples exhibit a PA of 0 across all models, which "could indicate not merely high difficulty but also potential flaws in task design." The authors are advised to conduct a case check to ensure the high quality of the benchmark.
>
> Thank you for highlighting this point. We have conducted a review of these cases. Detailed analyses and resolutions have been added to the revised manuscript, as described in Experiment section.

---

### Official Review · Reviewer_dHFc · 2024-10-30

**Soundness:** 3
**Presentation:** 2
**Contribution:** 1
**Rating:** 3
**Confidence:** 4

**Summary:**

The paper first proposes that the general task of reasoning can be broken down into sub-capabilities (e.g. path exploration, selection of relevant knowledge, multi-step inference). The paper then adjusts the focus on evaluating multi-step inference capability in isolation. To achieve such evaluation, the authors construct 23 tasks that require manipulation of strings, integers, and lists such that the exact procedure to complete the task is explicitly provided, minimal implicit knowledge is required, and is straightforward for humans. Although these tasks provide some insight into a model’s capability of performing rote tasks, it does not provide sufficiently interesting tasks that separates itself from other instruction-following benchmarks. In order to gather some insights regarding task complexity and measure a model’s ability to follow precise instructions, the authors develop some metrics, namely, prefix match length, prefix accuracy, sequential match, and final match. These metrics provide some insight into the role task complexity (i.e. increasing the number of steps required for the task) plays in a model’s ability to perform extended precise instruction following. However, the findings are that the final match accuracy provides similar information as the prefix accuracy and sequential match metrics, thus, ultimately providing limited additional insights into models’ capabilities.

**Strengths:**

- The paper introduces new metrics such as prefix match length, prefix accuracy, sequential match, and final match which provide some interesting analysis of the role task complexity (i.e. increasing the number of steps required for the task) plays in a model’s ability to perform extended precise instruction following
- The benchmark creation process is clear and is constructed in a manner to assess the capability coined as “instruction follow-ability”

**Weaknesses:**

- The 23 tasks created are not representative of real tasks that an LLM would perform in practice and there were no experiments that attempts to correlate the performance on this type of instruction followability task to general reasoning tasks
- The paper mentions that a novelty of their benchmark to other instruction following benchmarks is that they are able to assess the intermediate steps and not rely simply on the final outcome. However, their empirical findings show that FM is, in practice, similar to PA and SM, which begs the question of whether evaluating final outcome is already sufficient enough to assess step-by-step explicit instruction following.
- A lack of interesting ablations/error analysis. Some ideas could be to compare FM performance when the model is prompted to complete the set of steps in one go or error analysis on the types of mistake the model makes and a breakdown of which step (whether it is earlier or later steps) the model usually makes the wrong move. It would also be interesting to see if the mistakes are mostly just error in one step but the rest of the steps, conditioned on that error, are still accurate.

**Questions:**

- Is there some evidence that improving in this step-by-step instruction following where all the information is provided and the exact path is outlined is correlated with how well the LLM will be able to execute step-by-step instruction following when it needs to explore different paths + use implicit knowledge/world knowledge?
- Could you clarify how you transform the LLM output into a JSON format for assesement?
- Do you have a sense of how the model performance will vary if you don't prompt the model to do the task step by step but rather ask it to complete the steps in one go?

---

> ### Author Response · Authors · 2024-12-03
> **Response to Reviewer dHFc**
>
> > The 23 tasks created are not representative of real tasks that an LLM would perform in practice and there were no experiments that attempts to correlate the performance on this type of instruction followability task to general reasoning tasks
>
> The o1 series, particularly o1-preview and o1-mini, are generally recognized as models with strong reasoning capabilities. Our experimental results also confirmed that this series demonstrated superior performance compared to other models.
>
> Furthermore, the objective of this study is to clarify how this type of task correlates with the reasoning abilities of LLMs. While the 23 tasks do not directly represent real-world tasks, evaluating the ability to follow explicitly defined procedures reveals limitations of the models that existing benchmarks could not capture. Through this approach, we believe that it will become possible to quantitatively verify the correlation between instruction-following tasks and general reasoning tasks in the future.
>
>
> > The paper mentions that a novelty of their benchmark to other instruction following benchmarks is that they are able to assess the intermediate steps and not rely simply on the final outcome. However, their empirical findings show that FM is, in practice, similar to PA and SM, which begs the question of whether evaluating final outcome is already sufficient enough to assess step-by-step explicit instruction following.
>
> As shown in Figure 2, as the problem length (PL) increases, a tendency for performance to decline across all models is observed. This phenomenon is as expected, given that problems with longer procedures are considered more challenging for models. This observation serves as a fundamental validation that the benchmark structure has been appropriately designed.
>
> It should be noted that FM and PA appear to show similar trends within the range of shorter problems. Nevertheless, while FM focuses solely on the final outcome, PA evaluates intermediate steps as well, meaning their implications could diverge in the context of longer problems. An empirical analysis of the scores obtained from seven LLMs revealed that the correlation coefficient between PA and FM is high (up to 0.85) for shorter problems but decreases as the problem length increases. This result suggests that as problem length grows, the accuracy of intermediate steps becomes more critical, highlighting aspects that FM alone may fail to capture.
>
> Moreover, considering that SM (Sequential Match) and PA are stricter metrics than FM, the evaluation of intermediate steps becomes particularly important for exposing the weaknesses of models on longer problems. As illustrated in Figures 11 and 12, the distribution of PML (Prefix Match Length) for each model and task at various PL levels provides concrete evidence of the extent to which models can handle a given number of steps. This enables the extraction of detailed insights that cannot be obtained from FM alone.
>
> Therefore, the observation that FM is similar to PA or SM applies only to shorter problems. For longer problems, differences between these metrics become evident. The evaluation of intermediate steps, which is the novelty of this benchmark, is essential for detailed assessments of model performance, particularly in tasks involving lengthy procedures.
>
> > Could you clarify how you transform the LLM output into a JSON format for assesement?
> The transformation into JSON was performed using the Structured Outputs service provided by OpenAI, utilizing GPT-4o as the model. The Python API facilitates the conversion of text into JSON that adheres to a specified schema. As shown in Table 1, while the intermediate and final states of each task differ in type, these states can be extracted from the model's free-form responses by defining appropriate classes for each type. The system prompt used is provided in Figure~\ref{fig:structured_prompt}, where the model's predicted text is inserted accordingly.
>
> It is worth noting that the tasks in ProcBench inherently include the formatting of the model's output as part of the task itself. Consequently, this step should be regarded as a component of the model rather than as part of the evaluation function.

---

> ### Author Response · Authors · 2024-12-03
> **Response to Reviewer dHFc (2)**
>
> > Do you have a sense of how the model performance will vary if you don't prompt the model to do the task step by step but rather ask it to complete the steps in one go?
>
> We conducted the additional experiment to respond the reviewer comments. In the one-go setting, intermediate states were omitted, and prompts were modified to output only the final state. Nevertheless, the main procedures described in the `Template` were still provided. This setting evaluates the model’s performance when intermediate states are not strictly required.
>
> The one-go setting resulted in lower overall accuracy. Interestingly, the standard ProcBench setting, which includes intermediate states, demonstrated higher accuracy. These results suggest that the `Template` plays a role similar to Chain-of-Thought reasoning, supporting the step-by-step development of inference.

---

### Official Review · Reviewer_J9UW · 2024-11-04

**Soundness:** 2
**Presentation:** 2
**Contribution:** 1
**Rating:** 3
**Confidence:** 5

**Summary:**

The authors develop a new benchmark, ProcBench, to assess the capacity of big language models to follow multi-step instructions taken from library of 23 text based tasks. They tested the performance of seven mostly closed source language models on ProcBench. The models need to construct sequences of intermediate and final states. Further, authors come up straightforward metrics in order to assess the performance of models for these tasks.

**Strengths:**

- I believe, this paper is clearly written and easy to understand.
- Various metrics to measure instruction following capabilities

**Weaknesses:**

# Limitations


- Lack of Novelty : [1], [2] already demonstrate LLM's accuracy declines as, planning depth of procedurally generated problems increases. Paper doesn't propose any new technique that can improve/augment the instruction following capabilities.

- Insufficient Benchmarking: The paper fails to provide benchmarking insights across various model variants and sizes, which limits the usability of its benchmarking insights.

- Effect of tokenization : Given the tasks involve character manipulation, ablation about how different tokenization schemes (e.g., byte-level, subword-level, character-level) impact the models' ability to follow instructions and maintain coherence over long sequences would provide valuable insights into the relationship between input representation and procedural reasoning.

- Other techniques : techniques such as majority voting, self-consistency, self-refine [3] prompting to improve model adherence to instructions. Their analysis is completely absent.

[1] : Karthik V. et al “LLMs Still Can’t Plan; Can LRMs? A Preliminary Evaluation of OpenAI’s o1 on PlanBench,”

[2] : Kevin W. et al "On The Planning Abilities of OpenAI's o1 Models: Feasibility, Optimality, and Generalizability"

[3] : Aman N. et al "Self-Refine: Iterative Refinement with Self-Feedback"

**Questions:**

In addition to points stated in weaknesses, It would be good to include following analysis:

- Task Difficulty Analysis: In Figure 6, tasks are ordered by median accuracy without prior analysis or justification of what makes some tasks harder than others. A taxonomy based on task nature, required implicit knowledge, number of steps, etc., compared to empirical accuracy, would be informative. The paper mentions only 91 out of 5,520 examples had PA=0 across all models. Could you provide more details about these examples? Are there common patterns that make these particularly challenging?

- Robustness Analysis : Incorporating robustness checks such as perturbations in instructions or introducing minor distractions, It would add a practical dimension to ProcBench, aligning it more closely with real-world scenarios where instructions may not always be perfectly structured.

---

> ### Author Response · Authors · 2024-12-03
> **Response to Reviewer J9UW**
>
> > Lack of Novelty : [1], [2] already demonstrate LLM's accuracy declines as, planning depth of procedurally generated problems increases. Paper doesn't propose any new technique that can improve/augment the instruction following capabilities.
>
> Thank you for your comments. While we acknowledge that benchmarks such as PlanBench evaluate planning abilities through multi-step tasks, ProcBench focuses on a fundamentally different aspect of reasoning.
>
> ### **Focus on Execution Rather Than Planning**
> PlanBench evaluates a model's ability to generate plans for achieving a given goal. By its nature, it emphasizes planning capabilities. In contrast, ProcBench assumes that the plan (template) is already fixed and evaluates whether the model can faithfully follow the given plan and execute the steps correctly. This distinction is significant, as planning and faithfully executing a plan are distinct skills requiring different cognitive processes.
>
> For instance, designing a recipe and following the recipe as written are clearly different capabilities. ProcBench focuses on the latter—whether a model can execute a pre-determined plan accurately and consistently.
>
> ### **Complementary, Not Redundant**
> ProcBench and PlanBench may appear similar at first glance, but they are fundamentally complementary. While PlanBench evaluates plan generation, ProcBench assesses the ability to faithfully execute a given plan. This difference is also critical for understanding the diversity of human reasoning. Reasoning encompasses various forms, and categorizing and analyzing these forms are essential for advancing LLMs and AI systems in general.
>
> Evaluating complex tasks that require comprehensive reasoning is undoubtedly important. However, focusing on specific reasoning abilities is equally critical. Benchmarks such as PlanBench and ProcBench aim to measure distinct components of reasoning in isolation. This approach enables detailed feedback for model improvements and the design of new tasks.
>
> That said, we do not consider ProcBench sufficient on its own. Robust evaluation requires combining diverse tasks. To this end, ProcBench includes 23 types of tasks, but we anticipate further expansion of benchmarks and the development of new methods to evaluate reasoning in finer detail.
>
> ### **Insights into Broader Reasoning Challenges**
> ProcBench provides an essential perspective for deepening our understanding of reasoning capabilities. State-of-the-art LLMs demonstrate strong knowledge application in complex domains like physics and law. However, they often lack consistency when tasked with procedural adherence as evaluated in ProcBench. Even models considered strong in reasoning, like o1-preview, show improved performance but still fall short in maintaining consistency.
>
> This result highlights a limitation of current LLMs: while they excel in knowledge-intensive reasoning, they face challenges in multi-step procedural reasoning. Addressing this gap is crucial for applications requiring explainability and safety.
>
> Moreover, the challenges presented by ProcBench are notable in that they are straightforward for humans yet remain difficult for current AI models. Understanding this disparity points to future directions for developing more reliable and user-friendly AI systems.
>
> ### **Future Directions**
> This work is based on the hypothesis that reasoning can be decomposed into "planning" and "execution." While not all reasoning processes can be explained through this decomposition, this perspective is essential for a deeper understanding of reasoning. Human reasoning mechanisms are still not fully understood, and our hypothesis provides a foundation for designing AI systems capable of more reliable reasoning.
>
> ProcBench specifically focuses on "execution," providing a foundation for evaluating and verifying these elements individually or in combination. The stepwise evaluation metrics and task designs in ProcBench serve as valuable tools for diagnosing model weaknesses and driving further improvements.

---

> ### Author Response · Authors · 2024-12-03
> **Response to Reviewer J9UW (2)**
>
> > Insufficient Benchmarking: The paper fails to provide benchmarking insights across various model variants and sizes, which limits the usability of its benchmarking insights.
>
> In this paper, we evaluated some of the most prominent and highly regarded LLMs, particularly state-of-the-art (SOTA) models, providing a robust baseline for performance comparison. This allows us to clearly observe differences in performance across models, as demonstrated in the figures and tables, which reveal variations in metrics such as Prefix Accuracy and Sequential Match. This analysis serves as a solid starting point for evaluating the current state of LLM technology.
>
> Furthermore, even among the evaluated models, while they tend to succeed in shorter tasks, many fail to deliver consistent results in longer tasks. This highlights the difficulties current models face with multi-step reasoning, offering valuable insights into their capabilities and limitations. The significance of this research lies in providing a meaningful evaluation framework for future LLMs. Measuring the fundamental ability to handle multi-step reasoning tasks is expected to become increasingly important in future LLM research.
>
> On the other hand, we judged that evaluating smaller models or other variants at this stage would likely reveal many failure cases without yielding significant new insights. For this reason, we focused our analysis on SOTA models. Specifically, models such as o1-preview and o1-mini were included. While the details of these models are not fully disclosed, they can be regarded as representing the culmination of current techniques like Chain of Thought and advanced prompt engineering.
>
> That said, as noted in the reviewer’s comments, evaluating a broader range of models and different model sizes would indeed be an interesting avenue for future research. Additionally, given that our dataset is easy to generate, we anticipate that it will be possible to further expand the number of examples to enable more detailed evaluations of models in the future.
>
> Taking these points into account, we believe that this study not only provides sufficient insights into the current SOTA models but also establishes an important foundation for future LLM research.
>
> > Effect of tokenization : Given the tasks involve character manipulation, ablation about how different tokenization schemes (e.g., byte-level, subword-level, character-level) impact the models' ability to follow instructions and maintain coherence over long sequences would provide valuable insights into the relationship between input representation and procedural reasoning.
>
> In response to the reviewer’s comment, we conducted additional analysis to examine the impact of different tokenization methods on task accuracy. Specifically, we compared two tasks, "DeleteChar" and "DeleteWord," to explore how tokenization affects tasks with varying granularities of string manipulation.
>
> In the DeleteChar task, characters are sequentially removed from a random string, whereas in the DeleteWord task, words are sequentially removed from sentences extracted from the Wikipedia English corpus. These differences reflect distinct characteristics in how tokenizers handle the input. In DeleteWord, frequent words are often represented as single tokens, while other words span multiple tokens, leading to primarily token-level operations. Conversely, DeleteChar requires fine-grained character-level operations, which, particularly in random strings, often involve more intricate token manipulation and conversions.
>
> Our analysis hypothesized that DeleteChar would be more challenging for models due to its reliance on tokenization-related transformations. However, as shown in our results (Figure 6), there was no significant difference in accuracy between the two tasks across the models. Interestingly, DeleteChar demonstrated slightly higher accuracy, contrary to expectations. This result suggests that factors beyond tokenization, such as the model's internal representations, may play a critical role and warrant further investigation.
>
> While the extent to which tokenization impacts other tasks remains unclear, our findings suggest that its influence may not be as pronounced as initially anticipated. Instead, the nature of the tasks themselves and other considerations, such as task-specific properties, seem to have a greater effect on model performance.

---

> ### Author Response · Authors · 2024-12-03
> **Response to Reviewer J9UW (3)**
>
> > Task Difficulty Analysis: In Figure 6, tasks are ordered by median accuracy without prior analysis or justification of what makes some tasks harder than others. A taxonomy based on task nature, required implicit knowledge, number of steps, etc., compared to empirical accuracy, would be informative. The paper mentions only 91 out of 5,520 examples had PA=0 across all models. Could you provide more details about these examples? Are there common patterns that make these particularly challenging?
>
> In response to the reviewer’s comment, we conducted additional analyses on the relationship between task characteristics and model accuracy. The results are summarized below.
>
> First, we quantified the characteristics of each task by defining several variables. Specifically, we considered factors such as the number of characters in the template, the number of loops or conditional branches in the procedure, the data types of intermediate and final states (e.g., integers, strings, lists), whether the amount of information to be tracked increases as the procedure progresses (encoded as 1 if it increases, 0 otherwise), and so on.
>
> Next, we investigated the relationship between these characteristics and model accuracy (Prefix Accuracy, Sequential Match, Final Match) using multiple approaches. In addition to univariate and multivariate statistical analyses, we applied decision trees and random forests to evaluate the importance of each variable. Furthermore, the analysis was stratified by model and by task layer.
>
> However, the results showed no clear trends indicating that specific task characteristics significantly impact model accuracy. For example, tasks with longer templates or more conditional branches did not consistently exhibit lower accuracy, and no consistent relationships were identified.
>
> These findings suggest that it is challenging to establish a straightforward, predictable relationship between perceived task difficulty and LLM performance. Nevertheless, there remains room for future research to explore such correlations by generating more tasks and adding further data.

---

### Official Review · Reviewer_aoxV · 2024-11-06

**Soundness:** 3
**Presentation:** 3
**Contribution:** 3
**Rating:** 6
**Confidence:** 3

**Summary:**

The paper introduces ProcBench, a novel benchmark designed to evaluate the ability of large language models (LLMs) to accurately follow explicit, multi step instructions, a capability referred to as instruction followability. Unlike existing benchmarks that often require implicit knowledge or complex domain specific reasoning, ProcBench focuses on tasks where all necessary procedures and information are explicitly provided. This approach minimizes the reliance on implicit understanding and isolates the models' proficiency in procedural adherence.

ProcBench comprises 23 diverse tasks (e.g., DeleteChar, FillWord, Sort) that vary in complexity from short to long steps, emphasizing step-by-step correctness. The authors evaluate several state-of-the-art LLMs, including GPT-4 variants, Claude-3.5-Sonnet, Mistral-large, and Gemini-1.5-Pro, using newly introduced evaluation metrics: Prefix Accuracy (PA), Sequential Match (SM), and Final Match (FM). These metrics assess both intermediate and final outputs of multi-step sequences. The results reveal that while some models perform well on simpler tasks, their performance degrades significantly as task complexity and as the number of steps increase, highlighting current limitations in handling complex, multi-step procedural reasoning.

**Strengths:**

Novel Focus on Procedural Reasoning: The paper addresses a critical but underexplored aspect of LLM evaluation by isolating the ability to follow explicit, multi-step instructions without relying on implicit knowledge. This focus fills an important gap in existing benchmarks.

Comprehensive and Carefully Designed Controllable Benchmark: ProcBench includes 23 distinct tasks that cover a range of procedural challenges across different domains, involving string manipulation, list operations, and arithmetic computations. The tasks vary in length and complexity, providing a thorough and controllable assessment of models' capabilities.

Evaluation Metrics like Prefix Accuracy (PA) and Sequential Match (SM) capte not only the correctness of the final outcome but also adherence to each intermediate step.

Insightful Analysis: The study reveals significant performance drops in models as task complexity increases highlighting limitations of current LLMs and areas for future improvement.

Open-Source Dataset and Code Availability: Making ProcBench and the evaluation code publicly available promotes transparency, reproducibility, and encourages further research in this area.

**Weaknesses:**

Limited Real-World Applicability: While effective for isolating procedural reasoning, the tasks may not fully capture the complexities and nuances of real-world scenarios where implicit knowledge and domain-specific understanding are often required.

Focus on Specific Task Types: The benchmark predominantly includes tasks involving string manipulation, list operations, and basic arithmetic. This may limit the assessment of models' abilities in other types of procedures, such as complex logical reasoning, spatial reasoning, or tasks requiring hierarchical planning.

Insufficient Exploration of Prompting Strategies and Model Adaptability: The impact of different prompting techniques, chain-of-thought reasoning, or model fine-tuning on performance is not thoroughly investigated.

Insufficient explanation of causes of regression. It would be useful for the authors to dig into the actual model outputs and categorize failure cases (tokenization issue, format mismatch vs actual instruction followability issue).

**Questions:**

Exploration of Prompting Strategies and Model Fine-Tuning: Have you considered evaluating the impact of different prompting techniques, chain-of-thought reasoning, or fine-tuning strategies on model performance in ProcBench tasks?

Failure cases: Could you dig deeper into what are the failure cases of the models and what proportion of them are non-reasoning related (tokenization and formatting issues)?

---

> ### Author Response · Authors · 2024-12-03
> **Response to Reviewer aoxV**
>
> > Limited Real-World Applicability: While effective for isolating procedural reasoning, the tasks may not fully capture the complexities and nuances of real-world scenarios where implicit knowledge and domain-specific understanding are often required.
>
> We appreciate the reviewer’s observation regarding the distinction between real-world complexity and the procedural reasoning tasks evaluated in ProcBench. It is indeed accurate that ProcBench does not aim to fully replicate the nuances of real-world scenarios. This is, however, an intentional design choice. ProcBench focuses on isolating and evaluating procedural reasoning capabilities in a controlled manner, free from the confounding factors of implicit knowledge or domain-specific requirements. This design allows us to identify and address fundamental limitations in models that might otherwise remain obscured in more complex real-world settings.
>
> Addressing real-world scenarios often requires a wide array of skills, including domain-specific reasoning and implicit knowledge integration. However, evaluating such scenarios alone risks conflating procedural reasoning with other abilities, making it difficult to pinpoint core deficiencies. For instance, in many existing datasets, models can rely on knowledge reuse or patterns learned from training data to solve tasks without demonstrating robust procedural reasoning. ProcBench bridges this gap by providing a benchmark that explicitly targets the ability to follow instructions accurately and systematically, which is a critical foundation for broader generalization and adaptability.
>
> While ProcBench does not aim to directly replicate real-world tasks, it highlights procedural reasoning as a fundamental component necessary for real-world applicability. Consider, for example, a workplace scenario where file naming conventions, such as “date + project name + version number,” must be followed. This requires understanding and adhering to specific rules—e.g., “the date must follow the YYYY-MM-DD format, the project name must use an abbreviation, and the version number must be two digits.” Moreover, these conventions can vary across organizations, demanding a level of adaptability that depends on precise procedural reasoning. By systematically evaluating this core capability, ProcBench provides a foundation for models to develop broader real-world adaptability and reliability.
>
> Finally, while many recent large language models (LLMs) excel in tasks that rely on implicit knowledge, they often lack consistency in procedural reasoning. ProcBench tasks such as DeleteChar and Sort reveal that even simple, repetitive procedures that humans execute easily can lead to failures in models, particularly in step-by-step execution. This inability is not due to a lack of knowledge or context but rather to fundamental limitations in managing and following procedures. These findings emphasize the importance of improving procedural reasoning as a distinct capability. Strengthening this foundation will enable AI systems to better adapt to real-world scenarios that demand reliability, precision, and contextual flexibility.

---

> ### Author Response · Authors · 2024-12-03
> **Response to Reviewer aoxV (2)**
>
> > Focus on Specific Task Types: The benchmark predominantly includes tasks involving string manipulation, list operations, and basic arithmetic. This may limit the assessment of models' abilities in other types of procedures, such as complex logical reasoning, spatial reasoning, or tasks requiring hierarchical planning.
>
> ProcBench is a simple dataset designed to evaluate fundamental abilities in reasoning tasks. This design is not intended to directly solve complex challenges but rather to serve as a litmus test for assessing whether a model possesses the basic procedural reasoning capabilities required for more advanced tasks. For example, in the field of image recognition, datasets like "CIFAR-10" and "MNIST" have provided a foundational benchmark for measuring baseline capabilities before addressing more intricate tasks. Similarly, ProcBench has the potential to play a comparable role in the reasoning domain.
>
> The decision to focus ProcBench on text-based tasks reflects a realistic and effective choice, considering the current state of large language models (LLMs), which are specifically optimized for text processing. Text as a modality is both intuitive and practical for evaluating procedural reasoning, as it allows for straightforward assessment of a model's ability to follow instructions and execute procedures accurately. That said, procedural reasoning and reasoning more broadly need not be limited to the text modality. For example, formats like puzzles or board games (such as chess or Go), which involve visual, spatial, or strategic elements, are equally valid domains for reasoning. Developing datasets that encompass such modalities represents an important next step in evaluating more comprehensive reasoning abilities.
>
> With such future extensions in mind, ProcBench adopts a text-based and simple design as a foundational starting point. This enables easy comparison across models and facilitates the evaluation of basic procedural reasoning skills. Going forward, datasets incorporating visual data for spatial reasoning or physical modalities for tasks involving object manipulation and action planning should be developed to address more complex challenges. Through this phased evolution, ProcBench can provide a new framework for evaluating reasoning capabilities and serve as a basis for advancing research in the next generation of AI.
>
>
> > Insufficient Exploration of Prompting Strategies and Model Adaptability: The impact of different prompting techniques, chain-of-thought reasoning, or model fine-tuning on performance is not thoroughly investigated.
>
>
> ## Additional Experiments Based on Reviewer Comments
>
> Based on the reviewer comments, we conducted additional experiments to analyze the diversity of prompt strategies and the error tendencies in model performance for ProcBench tasks. These experiments were carried out using four models provided by OpenAI. For each problem length, we sampled 5 out of 10 questions and evaluated half of the dataset. While the scope of the experiments was limited, the results provided meaningful insights into how prompt strategies impact model performance.
>
> ## Impact of Few-shot Prompts
> In the few-shot setting, three examples were inserted between the `Template` and the `Question`. Each example consisted of a `Question` and its corresponding `Intermediate States` and `Final State`, presented as pairs. This setup was hypothesized to complement ambiguous natural language instructions and support complex outputs.
>
> ### Results
> Few-shot prompts improved accuracy across many tasks. Notably, the o1-preview model achieved very high accuracy in certain tasks. These results suggest that providing concrete examples reduces ambiguity and effectively aids the model in interpreting and applying instructions.
>
> ## One-go Setting
> In the one-go setting, intermediate states were omitted, and prompts were modified to output only the final state. Nevertheless, the main procedures described in the `Template` were still provided. This setting evaluates the model’s performance when intermediate states are not strictly required.
>
> ### Results
> The one-go setting resulted in lower overall accuracy. Interestingly, the standard ProcBench setting, which includes intermediate states, demonstrated higher accuracy. These results suggest that the `Template` plays a role similar to Chain-of-Thought reasoning, supporting the step-by-step development of inference.

---

> ### Author Response · Authors · 2024-12-03
> **Response to Reviewer aoxV (3)**
>
> > Insufficient explanation of causes of regression. It would be useful for the authors to dig into the actual model outputs and categorize failure cases (tokenization issue, format mismatch vs actual instruction followability issue).
>
> As part of the error analysis, we added a focused examination of cases where PA=0. This analysis revealed that failure cases primarily occur in the FillWord and MoveCyclic tasks. These specific examples are discussed in detail in the main text.
>
> Regarding tokenization issues, we considered that a direct comparison of different tokenization methods (e.g., byte-level or subword-level) is challenging due to model constraints. Instead, we addressed this issue by comparing the performance of similar tasks, which allowed us to infer the potential impact of tokenization on the results. This discussion has been added to the main text to provide further insights into the influence of tokenization.
>
> > Exploration of Prompting Strategies and Model Fine-Tuning: Have you considered evaluating the impact of different prompting techniques, chain-of-thought reasoning, or fine-tuning strategies on model performance in ProcBench tasks?
>
> In the few-shot learning experiment, we provided few-shot examples within the prompt, allowing the model to refer to them while solving the tasks. This approach generally improved the model's accuracy.
>
> Additionally, in the one-go setting, we modified the template to omit the requirement for intermediate state outputs, evaluating accuracy based solely on the final output. While we expected this simplification to enhance the accuracy of the final output, the results did not confirm such improvement. On the contrary, the explicit requirement for intermediate states appears to play a role in enhancing the model's overall accuracy.
>
> These findings have been added to the main text, providing further insights into how different prompting strategies and task designs impact model performance.

---

### Author Response · Authors · 2024-12-03
**General Response**

To all reviewers,

Thank you very much for taking the time to review our work and for providing many valuable critiques. Some of the feedback overlapped, so we are addressing the general points collectively here. We have added the following elements to the paper. For other points, we have responded individually.

- Additional experiments on Few-shot Learning as a major Prompt Engineering method
- Experiments where models are not required to predict intermediate states ("One-go experiment")
- Error analysis of the 91 cases with PA=0
- Analysis of the indirect impact of tokenization
- Details on Structured Outputs

Thank you again for your thoughtful feedback.

---

### Meta-Review · Area_Chair_QCtm · 2024-12-18

**Metareview:**

The paper proposes a multi-step instruction following benchmark for LLMs that mainly consist of  tasks that do not assume access implicit knowledge (e.g., string manipulation). The paper also argued to use evaluation metrics that capture not only the correctness of the final outcome but also adherence to each intermediate step. Similar to prior literature, the results show that performance of LLMs (include the recent state of the models such as o1, claude and Gemini) degrade as we increase task complexity and number of steps.

Strengths:

- Precise instruction following using procedural benchmarks and partial correctness metrics are quite useful for analysis and benchmarking beyond final accuracy.
- Open-Source Dataset and Code Availability for ProcBench  promotes reproducibility, and encourages future use.
- Clear presentation and well-written paper.

Weaknesses:
- Narrow set of tasks, focusing mainly on algorithmic and string manipulation. While such tasks are useful as diagnostics and surfacing differences in LLM capabilities, it's unclear whether higher performance on ProcBench corresponds to better reasoning capability on realistic tasks and common use cases of LLMs.
- Lack of ablations analyzing design choices, particularly tokenization schemes (e.g., byte-level, subword-level, character-level). Other minor improvements could incorporate  prompting techniques to make tasks more self-contained, such as many-shot ICL or very long instruction prompts (e.g., task tutorial).
- Lack of attribution analysis for current failure cases.  Fro example, planning and plan execution are not done independently, so it needs to be established how performance on ProcBench tasks relates to performance on the same tasks without explicit instructions to follow the specified plans (and just asking it to think step by step). The reviewers also suggested several interesting analysis / ablations.

Reasons to reject: While the paper focuses on benchmarking an underexplored aspect of LLM instruction following capability, the paper has major weakness due to lack of real world applicability and the toy nature of the benchmark. The discussion and authors engagement during rebuttal did improve my view of this work, but I'm still leaning towards rejection.

**Additional Comments On Reviewer Discussion:**

Most of the reviewers pointed several weaknesses, which the authors tried to address through their response. However, only one of the reviewers engaged and acknowledged that their concerns were not addressed. For other reviewers, I read the entire discussion and was not convinced that the pointed weaknesses were fully addressed, particularly about relating the utility of ProcBench to real world tasks and attribution analysis (pointed out by several reviewers). Overall, the paper seems to be making a step in the right direction, but changing the emphasis of the paper and addressing the weaknesses would make for a stronger submission.

---

### Decision · Program_Chairs · 2025-01-22

Reject